# Optimized Clustering Algorithms for Large Wireless Sensor Networks: A Review

**DOI:** 10.3390/s19020322

**Published:** 2019-01-15

**Authors:** Damien Wohwe Sambo, Blaise Omer Yenke, Anna Förster, Paul Dayang

**Affiliations:** 1Faculty of Science, University of Ngaoundéré, 454 Ngaoundéré, Cameroon; piusday@gmail.com; 2LASE Laboratory, University of Ngaoundéré, 454 Ngaoundéré, Cameroon; byenke@yahoo.com; 3ComNets, University of Bremen, 28334 Bremen, Germany; anna.foerster@comnets.uni-bremen.de

**Keywords:** large wireless sensor networks, clustering, metaheuristic, computational intelligence, machine learning

## Abstract

During the past few years, Wireless Sensor Networks (WSNs) have become widely used due to their large amount of applications. The use of WSNs is an imperative necessity for future revolutionary areas like ecological fields or smart cities in which more than hundreds or thousands of sensor nodes are deployed. In those large scale WSNs, hierarchical approaches improve the performance of the network and increase its lifetime. Hierarchy inside a WSN consists in cutting the whole network into sub-networks called clusters which are led by Cluster Heads. In spite of the advantages of the clustering on large WSNs, it remains a non-deterministic polynomial hard problem which is not solved efficiently by traditional clustering. The recent researches conducted on Machine Learning, Computational Intelligence, and WSNs bring out the optimized clustering algorithms for WSNs. These kinds of clustering are based on environmental behaviors and outperform the traditional clustering algorithms. However, due to the diversity of WSN applications, the choice of an appropriate paradigm for a clustering solution remains a problem. In this paper, we conduct a wide review of proposed optimized clustering solutions nowadays. In order to evaluate them, we consider 10 parameters. Based on these parameters, we propose a comparison of these optimized clustering approaches. From the analysis, we observe that centralized clustering solutions based on the Swarm Intelligence paradigm are more adapted for applications with low energy consumption, high data delivery rate, or high scalability than algorithms based on the other presented paradigms. Moreover, when an application does not need a large amount of nodes within a field, the Fuzzy Logic based solution are suitable.

## 1. Introduction

### 1.1. Background

Smart technologies are widely used in fields like building, health, ecological monitoring, security, home, vehicles, planes, or shipboard. However, smart environments rely first on sensory data from the real world like it is done by sentient organisms. Smart environments are possible due to the recent evolution of wireless communication technologies, digital electronics, and MEMS technology which have seen the apparition of sensors. They are small in size and are able to collect information on its environment like temperature, pressure, humidity, water content, gas presence, or luminosity. In spite of the large amount of applications offered by WSN, sensor nodes are designed with resources constraints such as a restricted computing capacity, reduced memory size and storage, weak range of communication, low bandwidth, and a limited amount of energy. To efficiently cover areas, a single sensor is not sufficient due to its limited communication range. In order to cover a more consequent space, several sensors are deployed and connected to each other, thereby forming a Wireless Sensor Network (WSN) [1].

By designing a WSN, the most important parameter to be taken into account is the energy consumption because it defines the lifetime of a sensor node and thereafter the lifetime of the whole network [2]. A trade-off between the energy restriction and the resources constraints of sensors is imperative for the performance of the network. Nevertheless, when the amount of nodes within the network is extensive, the usual direct routing consumes more energy and may reduce the network lifetime considerably [3]. From the original wireline networks, hierarchical or cluster-based routing are widely used for large WSNs because they are techniques with advantages related to scalability, efficient communication, and fault tolerance [4]. In hierarchical architectures, the whole network is divided into sub-networks called clusters. Each cluster is led by a special node named Cluster Head (CH) which is responsible for gathering or fusion data from nodes that belong to the same cluster [5]. In this kind of routing technique, the inter-cluster and intra-cluster communications may act in a multihop manner. Thus, a sensor node communicates only with its nearest neighbor in order to preserve its remaining energy and not to waste its energy by trying to communicate with a neighbor which is far away [4]. Other clustering approaches aim at finding a trade-off between the reliability of sensing and communication overhead based on unsupervised learning process [6].

Meanwhile, traditional hierarchical based algorithms are mostly probabilistic clustering approaches [7]. However, like the authors of [8], for example, the clustering on WSN remains a Non-deterministic Polynomial (NP) hard optimization problem which cannot be solved efficiently by traditional approaches. Thus, a more accurate resolution of the NP optimization of the clustering is made possible by using approaches based on recent research on Computational Intelligence (CI) and Machine Learning (ML). These recent clustering paradigms are known as optimized clustering algorithms.

The clustering solutions based on CI or ML consider environmental and biological behaviors and outperform most of the traditional clustering solutions in terms of scalability, reliability, fault tolerance, amount of data delivered, energy consumption, better coverage of the experimental field, and the increase of the network lifetime [7,8,9,10,11]. Meanwhile, according to the type of application, the choice of the appropriate approach is very important both on the cost of the network deployment and its lifetime.

### 1.2. Author’s Contributions

The main goal of this paper is to give a quick overview to a beginner or senior in the research field about clustering approaches based on ML and CI, it also helps readers in choosing a clustering technique adapted to the specifications of their application. To achieve it, we conducted a large review of the recent optimized clustering solutions. They are classified by the ML/CI implemented; thus, several performance parameters are used to compare and to evaluate them. It is then easy to have an overview of MI/CI based approaches according to the requirements of WSN’s applications. In short, our main contributions can be summarized as follows:Wide review of recent intelligent clustering approaches for WSN;Classification of algorithms in terms of the CI or ML used;Intensive evaluation and comparison of the selected algorithms according to 10 parameters.

### 1.3. Paper’s Organization

The rest of this paper is organized as follows: Section 2 shows the principal routing techniques used in WSN; in Section 3, a wide presentation of the optimized clustering based on ML and CI are given; a comparison between several optimized clustering approaches is provided in Section 4; conclusion and directions for future work are presented in Section 5.

## 2. Routing on Wireless Sensor Networks

The main goal of routing in WSN is to carry out data communication when trying at the same time to prolong the network lifetime and provide high quality of service during data delivery [12]. Based on the network structure, routing on WSN can be classified as data centric based routing, location based routing, group-based routing, or as hierarchical based routing.

### 2.1. Data Centric Based

In several sensor networks, it is not obvious to assign an identifier to each node because of the large number of nodes deployed. Besides the problem of identification of nodes, the random deployment of nodes makes it difficult to select a specific node during the routing of data through the network. Nevertheless, since data are usually transmitted from each node within the deployment region, the redundancy of these data can be significant then waste a lot of energy. 

The resolution of redundant data during routing has led to the data centric approach, which is different from the traditional address-based routing where routes are created between addressable nodes [12,13,14]. In data centric based routing, before data have to be sent by nodes in a selected region, the sink node should send queries to a selected region and wait for the incoming data [12]. The first and the most popular data centric protocol is the Sensor Protocols for Information via Negotiation (SPIN), in which negotiation between nodes is considered in order to eliminate redundant data and reduce the energy consumption. There are several kinds of data centric based routing like Directed Diffusion [15], Energy-aware routing, Rumor routing, or Gradient-Based Routing [13,15].

### 2.2. Location Based Routing

The position of sensor node is required in applications like military tracking, ecology monitoring, or health care. Contrary to data centric based routing where the position of a node can be unknown, the location-based protocols are very interesting since they can significantly decrease the complexity of finding best routes through the network. The distance between two neighbor nodes can be therefore estimated by the Received Signal Strength (RSS) [12]. When the study area is well known in advance, using the location of sensors will eliminate the number of transmissions significantly because the queries would be assigned only to a particular region at a particular time [12]. However, information about a position can be done through the use of a GPS (Global Positioning System) module on the sensor. Since the one goal when designing a sensor network is low cost and energy management, the use of GPS by sensors on a large scale network is quite expensive and energy consuming [16]. An example of location-based protocol is the Minimum Energy Communication Network (MECN), it reduces the energy consumption into the network by using a low power GPS module on each sensor node. Meanwhile, it is best applicable to sensor networks, which are not mobile [12]. Another well-known location-based algorithm is the Geographic Adaptive Fidelity (GAF) designed initially for mobile ad hoc networks. Presentations of location-based protocols are conducted in [3,17].

### 2.3. Group Based Routing

Having the location of each node within a field is not easy when the amount of sensors increases considerably. A more easy approach consists of deploying sensor nodes in groups. In this kind of routing, nodes in the same group are most of the time closed to each other [18,19]. In group-based routing solutions, each group is able to perform its own application independently. Take, for example, the measurement of the environmental impact of an area made up of a small forest, a sandy place, and a marine reef. In this example, three groups can be deployed according to the measurements of each application (forest, sand, underwater). However, the deployment of group-based protocol within the study field needs to be meticulous. As explained by Donggang et al. [19], in a group-based solution, each sensor node is assigned to its group before the deployment. There are several algorithms based on group routing. Lloret et al. [18] proposed the Group-Based Protocol for Large Wireless Ad Hoc and Sensor Networks called GBP-WAHSN. Another group-based algorithm is called Group based Mobile Agent Routing (GMAR) [20], which uses a mobile agent in order to aggregate data in each group.

### 2.4. Hierarchical Based Routing

Clustering is an efficient topology control approach for maximizing the lifetime and scalability of WSNs. The hierarchical based routing is a part of the group-based routing and consists of creating a virtual hierarchy among the nodes of the sensor network [21]. This class of routing techniques is generally designed for large scale networks and aims to efficiently maintain the energy consumption of sensor nodes and increase the network lifetime by cutting the whole network into clusters [14]. Each cluster is led by a node called Cluster Head (CH) which receives data from nodes within the cluster. CHs communicate each other in order to find a better route up to the sink node or the BS. This is done in order to reduce the energy consumption of sensor nodes by reducing the number of transmitted/received messages to the sink node. In addition to CH election, a second special node called Vice Cluster Head (VCH) can be elected in order to improve the lifetime of the CH as shown in [22]. Mechanisms like multihop communication, data aggregation, and data fusion are performed so that the energy is efficiently used within the cluster [12,13]. The most popular clustering algorithm is the Low-Energy Adaptive Clustering Hierarchy (LEACH), it uses probability computing and the received signal strengths to locally select the CHs which have to serve as router of the data up to the BS. In LEACH, local data fusion and aggregation are performed by local CH [3]. For a large scale network, LEACH is able to increase the network lifetime [21]. However, due to its single hop configuration, the CH on LEACH is assumed to have a long communication range. Thus, the data sent by the CH has to reach the BS directly. Another approach subdivides the problem into two layers: an organization layer to manage communications and a distribution layer made up of cluster members [23]. Many hierarchical based routing algorithms are proposed in the literature, such as the PEGASIS, TEEN, EEHC, PEACH, or HEED. Authors of [4,6,16,18,19] present several classical hierarchical algorithms and show how the scalability, the energy efficiency, network lifetime, data delivery, and the fault tolerance are greatly improved on large scale sensor network.

Except for the network structure, protocols in WSN field can also be classified according to the path establishment (proactive, reactive, hybrid); the protocol operations (multipath-based, query-based, negotiation-based, delivery-based, QoS-based, coherent-based); the next hop selection (broadcast, location, comment, probabilistic). Figure 1 gives a global overview of this classification. More details about each category can be found in [4,24,25,26,27].

## 3. Optimized Hierarchical Based Routing Protocols

The optimization problem of traditional hierarchical based routing has conducted to intelligent clustering strategies commonly known as optimized hierarchical based protocols [4]. The optimized clustering strategies are more recent and aim to be intelligent algorithms which improve the lifetime of a sensor network and make them energy efficient. For that, they are based on a recent development of ML/CI which is defined by [8] as an intelligent computational methodology that uses heuristic algorithms to obtain approximate solutions to NP hard problems efficiently. Several ML/CI paradigms used in clustering in WSNs can be classified as follows: Fuzzy Logic, Genetic Algorithm, Neural Network, Reinforcement Learning, and Swarm Intelligence paradigms [7,8,20,21,28,29,30]. Figure 2 presents the classification of optimized clustering algorithm for WSNs.

### 3.1. Fuzzy Logic

The Fuzzy Logic (FL) is a mathematical discipline invented to express approximate human reasoning. Contrary to the classical set theory which enable elements to belong or not to a set, FL allows a measure of imprecision or uncertainly which is marked by the use of linguistic variables like most, many, frequently through rules within a set called fuzzy set [9]. An example of a fuzzy set used for input variables of the distance between a node and the BS is presented in Figure 3. From the figure, the distance to BS is classified in close, medium, or far. Many works on optimized clustering based on FL are conducted in literature.

#### 3.1.1. FCH

The Cluster-head Election using Fuzzy Logic (FCH) is presented in [31]. It integrates a FL approach for the election of the CH based on the energy, node concentration, and centrality during computation. CH is elected by the BS in each round by defining the chance each node has to become the CH, helped by the fuzzy descriptors (energy, concentration, and centrality). The model of FL consists in four steps: fuzzification for the input descriptors, rule evaluation, aggregation of rules, and the defuzzification. LCH is compared to the famous LEACH protocol, it ameliorates the FND better than in LEACH and then increases the network lifetime in relation to LEACH.

#### 3.1.2. CHEF

Kim et al. [32] proposed the Cluster Head Election mechanism based on the Fuzzy logic (CHEF). Contrary to LEACH, CHEF uses the distance to the BS and the remaining energy of node for the CH selection. At each round, a node generates a random value between 0 and 1, and thereafter compares the obtained value with a threshold, Popt. If the random value is smaller than Popt, the chance value is calculated by using fuzzy IF-THEN rules. Thus, the node sends a message called Candidate_Message with the calculated chance. The Candidate_Message sent by the node means that it is a candidate for being a CH with the value of its chance. The node which sent a Candidate Message, waits for Candidate Messages and appropriate chance values from its neighbors. If its own chance is bigger than every *chance* from other nodes, the sensor node notifies the network with a CH-Message which means that the sensor node itself is elected as the CH. If a node which is not a CH receives the *CH*-message, the node selects the closest CH as its CH and joins a cluster by sending a Cluster Join Message to the CH. For the evaluation of CHEF, the authors consider 400 nodes randomly distributed in the area and the BS is not located within the experimental field. The comparison between LEACH and CHEF has shown that the cluster formation on CHEF was more efficient than on LEACH. Thus, CHs are not as closed with the proposed algorithm as with LEACH; there are too many nodes within a cluster that may rapidly decrease the energy of the CH. The results of experimentations have also shown that the lifetime of the network on CHEF is 22.7%, which is better than the lifetime of the network with LEACH according to the FND.

#### 3.1.3. LEACH-FL

LEACH-FL is an improved version of LEACH using FL [33]. It has a similar mechanism with LCH but the variables (parameters) used are battery level, distance from sink, and the node density. Election of CH is centralized on the sink, which has to calculate the chance of nodes becoming the CH like in LCH. LEACH-FL has three main parts: fuzzification module (four functions), an inference engine, and a defuzzification module. The authors of [33] compare LEACH-FL and LEACH, the experimentations have shown that the proposed algorithm has a lower energy consumption rate than LEACH. The network lifetime of the network using LEACH-FL outperforms the network lifetime when using LEACH.

#### 3.1.4. ICT2TSK

ICT2TSK is an improved clustering algorithm which uses a type-2 Takagi-Sugeno-Kang (TSK) as FL system [34]. ICT2TSK is used to elect the CH and to choose the one which can deal with the rule uncertainties better than a type-1 TSK FL system. It balances the network load by introducing a fixed competition radius for each CH so that energy is improved efficiency. When the network starts to work, all nodes use the LEACH protocol to send information about their location and their residual energy. ICT2TSK is centralized and uses the BS to select CHs. The FL system ICT2TSK is used to calculate the probability for each node to become a CH according to its residual energy, distance to the BS, and number of neighbor nodes.

#### 3.1.5. SEP-FL

SEP-FL is a FL approach improving election of CH within heterogeneous WSN [35], it is an improved version of SEP which is based on the election of CH by balancing the probabilities on the residual energy for each node. SEP-FL provides a longer stability period and a lower instability period and increases the lifetime of nodes. The approach is based on the distance from the BS and the residual energy level of each node type. The fuzzy system is divided into two Fuzzy Inference Systems (FIS); one for each type of nodes (advanced and normal nodes). SEP-FL is compared with LEACH, LEACH-LF, and the original algorithm SEP, the results show that SEP-FL increases the network lifetime and reduces the energy consumption better than the other three algorithms.

#### 3.1.6. EAUCF

Authors of [36] have proposed a fuzzy based algorithm for the energy-aware in unequal clustering called EAUCF. It aims to reduce the energy consumption of CHs within clusters since they are either close to the BS or may have low remaining battery power. The FL implemented in EAUCF uses an *if-then* mapping rules to handle doubts in CH radius estimation. Moreover, the algorithm also uses a probabilistic model which is employed for electing tentative CHs. In each clustering round, every sensor node has to generate a random number between 0 and 1 and compares it with the predefined threshold. If the random number is more than the threshold, the node becomes a tentative CH, for electing the final CH, EAUCF take into account the residual energy and the distance to the BS before calculating the competition radius. EAUCF has a better performance compared to LEACH, CHEF, and EEUC in terms of the death of first node, the half nodes alive, and the energy consumption.

#### 3.1.7. DFLC

DFLC (Distributed Fuzzy Logic-Based Clustering) is a clustering algorithm based on the fuzzy logic which is executed on a distributed manner by nodes within the network [37]. DFLC considers the network as a tree in which a node can be sink (BS), root (CH), member (node within a cluster), parent or child node. A parent node is an intermediate node and receives data sent by a child node to the root. In order to efficiently select the CH, each node runs the fuzzy logic engine with five input parameters: the residual energy, the distance of a node to the other nodes in the tree (centrality), the distance to the BS, the number of hops, and the number of neighbor nodes of a node (node density). During the execution of the fuzzy logic engine only necessary nodes that have a higher probability of being selected as a new root node are considered. DFLC avoids the crash of the network when sensor nodes may not perform their responsibilities due to the energy depletion. When a node receives a Discovery message during the set-up phase, it stores the id of the sender node and denotes this node as a closest neighbor node to the sink node, thus if the network crashes, an alternative path can be used to adapt the system to failures without delays. DFLC is implemented in NS2 simulator to test and compare its performances in terms of the energy consumption, the number of nodes alive, the network lifetime, and the amount of received messages within five networks (100, 200, 300, 400, and 500 nodes). The proposed algorithm is compared to LEACH, ACAWT [38], FCH, and CHEF. The results of experimentations shown that DFLC outperforms the others algorithms in terms of all the experimental metrics.

#### 3.1.8. SIF

SIF is a Swarm Intelligence protocol based on FL routing and considers the residual energy [39], the distance to the sink, and the distance from the cluster center to select appropriate CHs. SIF uses a Fuzzy C-Means (FCM) clustering algorithm to cluster all sensor nodes into balanced clusters, the appropriate CHs are selected via the Mamdani fuzzy inference system. SIF integrates a hybrid swarm intelligence called FA-SA. FA-SA is based on the firefly behaviors algorithm (FA) and the powerful local search algorithm SA, both are used to optimize the fuzzy rule base table of the fuzzy system. Authors have shown that SIF is energy efficient in terms of forming balanced clusters, minimizing the intra-cluster distances, prolonging the network lifetime, and maximizing the total number of data packets received in the sink. The comparison of SIF against protocols LEACH, LEACH-DT, ASLPR, and LEACH-FL shows that the proposed protocol significantly increases the network lifetime (by increasing the first node dies FND, the half node die HND, and last node dies LND). The results obtained also show that the amount of data received by the sink node on SIF is better than the others protocols according to the FND, the HND, and the LND.

#### 3.1.9. FBUC

FBUC or Fuzzy Based Unequal Clustering is an improved version of EAUCF [40]. Additionally, to EAUCF, FBUC uses a probabilistic threshold value instead of a predefined threshold value and adds a fuzzy variable called node degree used for the election of the CH during competition of radius. The members join the CH based on distance and the CH degree in order to efficiently use the energy and then increase the network lifetime. FBUC is compared with protocols LEACH and EAUC in two scenarios, WSN#1 (sink node inside the sensor field) and WSN#2. FBUC shows a better energy consumption and a better network lifetime than the other two algorithms according to the FND and the LND on the different scenarios.

#### 3.1.10. EEDCF

The authors of [41] propose an energy-efficient distributed clustering algorithm based on fuzzy called EEDCF. The proposed approach defines four (04) different states for each node: the initial state, the competing CH state, the elected CH state, and the member node state. During the first phase, each node has to build its own information table which contains the node ID, its residual energy, the neighbors’ ID, and their corresponding remaining energy. After each round, the node updates its information helped by the packet Node_MSG, which classifies the proximity of neighbors from itself. At the second phase, the node performs the fuzzy logic analysis based on its residual energy, the number of neighbor’s nodes within its communication radius (node degree) and the average residual energy of its neighbors. At the end of this step, each node turns into compete CH state and sends its output to all neighbor nodes within its communication radius. Where the output is obtained through IF-THEN rules in accordance with the mechanism of the TSK fuzzy model, the node with the lesser output turns into the member node state, and waits for joining a suitable cluster after CH election. However, the node with higher output turns into the elected CH state as well. Based on the RSSI, nodes join appropriate CH and thereafter build clusters. In order to evaluate EEDCF, authors have considered two scenarios with the same area size and the same location of the BS. The first scenario deploys 100 nodes whereas the second scenario uses 150 nodes within the study field. The experimentations shown that EEDCF outperforms the distributed algorithm for the energy-efficient data gathering protocol based on cluster structure called EADEEG [42] and DFLC. The lifetime of the network on EEDCF is better than on EADEEFG and on DFLC and has the best value of FND, HND, and LND in both scenarios. The results have also shown that the proposed approach had a better data delivery rate than EADEEG and DFLC.

### 3.2. Genetic Algorithm

The Genetic Algorithm (GA) is an adaptive heuristic approach based on biological genetic evolution for intelligent search and optimization. GA models the natural evolution by performing fitness tests on new structures to choose the best population [9]. With GA approaches, a population is made up of a group of chromosomes where a chromosome represents a complete solution to a relevant problem, and fitness shows the quality of a chromosome in function of concrete needs [28]. This kind of optimized algorithm is used for randomized search and optimization during routing of data. GA showed flexibility in solving dynamic problems and has been successfully applied within many NP-hard problems which include the clustering problem on WSN [7,8,20]. An example of the GA model is presented in Figure 4. Some clustering approaches based on GA are presented below.

#### 3.2.1. Wazed et al. [43]

Authors of [43] have proposed a Genetic Algorithm in extending the lifetime of two-tiered sensor networks. It schedules the data gathering of relay nodes and can significantly extend the lifetime of the relay node. A relay node acts as a CH and receives data from nodes which belong to its own cluster. Each relay node has to transmit data either to a simple node or another relay node or the BS. The chromosome is represented here by a specific routing protocol as a string of the node numbers where the length of each chromosome is equal to the number of relay nodes. The proposed routing algorithm is compared to two models of routing: the traditional Multi-Hop Data Transmission Model (MHDTM) and the Minimum Transmission Energy Model (MTEM). MHDTM finds the optimal path until the BS, whereas in MTEM, each relay node *i* transmits to its nearest neighbor *j* if this last is closer to the BS than the relay node *i*. Applied on large scale network the proposed GA approach significantly extends the lifetime of the network than the other two models.

#### 3.2.2. Hussain et al. [44]

The proposed technique uses a GA to build an initial set of clusters. In the beginning, the proposed GA uses all nodes in order to create clusters; however, during computation only alive nodes are used to create clusters so that the early death of some nodes due to data transfer is prevented. This approach is centralized because the formation of clusters is done by the BS. The chromosome presented by Hussain et al. [44] is designed to minimize the energy consumption and increase the network lifetime. The parameters used here are the distance to the BS, the cluster distance, the energy level, and the data transfer. The algorithm proposed in [44] is compared to LEACH, HCR-1, and HCR-2. The results obtained show that the approach increases the network lifetime by improving the number of alive nodes than the other algorithms.

#### 3.2.3. LEACH-GA

LEACH-GA is an improved version of LEACH based on a GA [45]. In addition to the set-up and the steady-state of LEACH, LEACH-GA has a preparation phase where nodes initially perform the selection of the CH and determine whether or not each node should be a candidate CH (CCH). During the preparation phase, information like node status, IDs, and the location are sent until the BS. LEACH-GA is also a centralized protocol since the BS receives messages sent from all nodes and performs GA operations in order to determine the optimal arrangement which shall minimize the energy consumption in each round. The proposed algorithm determines the optimal threshold probability for the formation of a cluster. Results have shown that LEACH-GA improves the network lifetime than LEACH by increasing the number of nodes alive per round.

#### 3.2.4. GABEEC

GABEEC is a GA based for energy efficient clusters in WSN proposed in [46] in order to improve the network lifetime. Contrary to LEACH-GA, GABEEC has only two states: set-up and steady-state. In the first phase, all clusters are created statically one time, but CHs within a cluster are changing dynamically based on the residual energy. GABEEC uses a binary representation of the network where each sensor node is a bit in which “1” correspond to a CH and “0” to a simple node. An instance of the network represents a specific chromosome, the GA evaluates each chromosome and selects the best profile so that the energy is efficiently used then increases the network lifetime. GABEEC is compared at first to LEACH (100 homogenous nodes), afterwards to HCR and the algorithm proposed in [44] (200 nodes used). The results show that GABEEC has a better percentage of alive nodes in both simulations.

### 3.3. Neural Network

Neural Networks (NNs) are mathematical models inspired from biological networks of neurons. Similar to a large and dense network, each neuron is connected to many other neurons. A NN consists of a network of neurons organized in input, hidden, and output layers where the NN learns the different paths and determine their interrelationships [20,47]. Figure 5 below presents a simple model of NN. The NNs are used for solving the problems of sensor fusion, data mining, and clustering. Some NNs solutions for clustering are presented below.

#### 3.3.1. Cordina & Debono [48]

The algorithm proposed in [48] presents a novel approach of clustering based on Self Organizing Map (SOM) neural networks. It aims to optimize the network by cutting it efficiently into clusters so that the energy consumption is minimized and the power needed by each node is reduced. It performs mechanisms which include a minimum separation distance from the CH. A SOM strategy is used for CH election, a system of CH rotation and a load balancing of helps by cost functions. Each round starts with a set-up phase in which the clusters are organized. During the transmission phase, data from nodes are transferred or not to the CH depending on scenarios. Data received from nodes of a cluster by the CH are aggregated and relayed to the BS. The election of the CHs is carried out by SOM using a 4-input, 16-output neuron mapping, but nodes and CHs which have a residual energy less than critical energy, send a “*Node dead*”/”*CH dead*” message directly to the BS. The results of comparison have shown that the strategy presented in [48] increases the network lifetime (FND and HND is better) than the traditional LEACH. The presented protocol also has an amount of data delivered which is better than on LEACH.

#### 3.3.2. Kumar et al. [49]

The algorithm presented in [49] is a NN based for energy efficient clustering and routing, which aims to optimize the network lifetime. It uses an LP formulation where the objective is to choose a number of nodes with higher levels of residual energy to form an optimal route, while minimizing the total routing cost. The CH election is conducted by using adaptive learning in NN joined by routing and data transmission. Contrary to [48], the proposed strategy is defined in three phases: set-up, routing, and data transmission phase. The set-up phase allows the election of CHs taken into the set of CHs nodes based on a cost metric. During the routing and data transmission phase, the proposed protocol scans all routes (store in R) and finds the best route to reach the BS. A source will then select the route that is the most energy efficient so that it maximizes the network lifetime. The Strategy proposed in [49] is compared to PEACH and results show that the proposed algorithm performs better by increasing the percentage of alive nodes.

### 3.4. Reinforcement Learning

The Reinforcement Learning (RL) is a sub-domain of ML which teaches an agent on what to do and how to assign situations to particular actions so as to be intelligent [10]. The agent would try several actions, and learns from its experience the best action it has to choose in order to optimize the network performance [8,9,10,20]. Most of the RL based protocols are used in some clustering algorithms in WSN finding optimal paths and prolonging network lifetime [11].

#### 3.4.1. CLIQUE

CLIQUE is a role free clustering which use a RL algorithm called Q-Learning [50]. On the proposed algorithm, nodes should know the identity of the cluster to which they belong. Contrary to traditional clustering approaches, CLIQUE does not have a conventional phase to select CHs by nodes within a cluster, but rather uses a Q-Learning strategy. The RL used on CLIQUE allows nodes to learn and independently decide whether or not to act as a CH. CH would be the cluster node which has the best routing cost to all sinks. CLIQUE assigns Q-values to each possible action; the learning process allows the agent (node) to select and execute an action (routing) and receives afterwards the corresponding reward used to update the Q-value. CLIQUE is compared to the Traditional Random Cluster head Assignment (TRC) which uses an a-priori probability at each node to decide whether or not to become a CH. The results obtained after simulations have shown that CLIQUE has a better performance than TRC in terms of energy consumption (FND), data delivery, and communication overhead.

#### 3.4.2. Ramli et al. [51]

Authors of [51] propose a cooperative algorithm based on RL for energy efficient dual hop clustered networks. The major contribution of the proposed strategy is to allow cluster members to exchange channel historical information in order to facilitate learning. It allows each cluster to adapt the number of required preferred channel size depending on the local area density and traffic. The proposed cooperative RL assumes two phases: the exploration and exploitation phases. During the exploration phase, for each successful transfer by a cluster member, its CH updates the weight on a channel. Meanwhile, during the exploitation stage, a cluster member used a channel when it had a weight of more than the threshold value.

### 3.5. Swarm Intelligence

The Swarm Intelligence (SI) is defined in [52] as “any attempt to design algorithms or distributed problem-solving devices inspired by the collective behavior of social insects and other animal societies”. Most of the proposed are based on the social behaviors of flocks of birds, schools of fishes, and insect cooperation like ants, bees, butterflies, etc. which have limited resources like sensor nodes used in WSN. SI approaches can be classify into Particle Swarm, Ant Colony, and Bee Colony Optimizations [8,11,43].

#### 3.5.1. Particle Swarm Intelligence

The Particle Swarm Optimization (PSO) is an evolutionary computation technique and is related to the bird flocking, fishing schooling, and swarm theory. Like the other evolutionary computation techniques, PSO is a population-based search algorithm and is initialized with a population of random solutions, called particles. A particle will have a fitness value, which will be evaluated by a fitness function to be optimized in each generation [53,54]. In order to increase the performances of WSNs, clustering strategies using PSO are proposed nowadays.

##### PSO-C

An energy-aware clustering approach for WSN using PSO is proposed in [55]. The authors presented a distance based and centralized approach which takes into account the maximum distance between the non-cluster head node and its CH, and the remaining energy of CH candidates in the CH selection. In the beginning, during the setup phase, all nodes send information about their residual energy and positions to the BS. To ensure that only nodes with a sufficient energy are selected as CHs, the nodes with an energy more than the average are eligible to be a CCH for each round. The PSO algorithms are executed at the BS to determine the best k-CHs which can minimize the cost during routing. PSO-C is compared to the traditional clustering LEACH and its amelioration version LEACH-C on two different scenarios. In the first scenario, the BS is located within the sensor field, whereas in the second scenario the BS is located outside the sensor field. The results after simulations have shown that PSO-C improves the amount of data delivered to the BS to 101%, 25% better than LEACH and LEACH-C, respectively, for the first scenario and 60%, 50% for the second scenario. PSO-C also increases the network lifetime by improving the number of alive nodes in both scenario.

##### Kuila & Jana [56]

Authors propose an energy efficient clustering and routing algorithms for WSN based on the PSO paradigm. The routing algorithm is developed on an efficient particle encoding scheme with a trade-off between transmission distance and multi-objective fitness function. Meanwhile, the clustering algorithm is presented by considering energy conservation of the nodes through the load balancing strategy. At the beginning (network setup), the protocol performs three phases: bootstrapping (each gateway and node are assigned unique IDs); route setup; and the clustering phase. The gateways can collect the IDs of the sensor nodes and the other gateways that are within their communication range and finally send the information to the BS. All the CHs which are heavily used as next hop relay nodes in data forwarding are assigned a lesser number of sensor nodes. Thus the energy consumption of the CHs is significantly balanced and the lifetime of the network is improved. The algorithms are based on the derivation of efficient particle encoding scheme and fitness function for routing and clustering separately. The algorithm is evaluated using two network scenarios WSN#1 (BS is located outside the sensor field) and WSN#2 (BS is located in the center of the sensor field). The proposed algorithm is compared to GA-based approach and the two clustering strategies GLBCA and LDC. Results show that the proposed PSO-based approach increase significantly the network lifetime (First Gateway Die FGD and Last Gateway Die LGD) on both scenarios than the other three algorithms. Results have also shown that the PSO-based approach increases the data delivered to the BS and reduces the number of hops until the BS than the other strategies.

##### PSO-HC

The PSO-HC is a hierarchical clustering based on the PSO paradigm [57] to maximize the network lifetime by minimizing the number of active CHs and to maximize the network scalability by using two-hop communication between the sensor nodes and their respective CHs. PSO-HC has also two phases per round: the set-up and the steady phases. During the set-up phase, each node updates its neighbor table with the RSSI value and sends it to add to ID and residual energy until the BS using the flooding method. The BS finds the average energy level of all nodes and only nodes with an energy level above the average are eligible to be a CCH. After, the BS runs the PSO algorithm to find the best k-CHs where a particle is represented as a sequence of candidate CHs ID’s. For the formation of the cluster, the BS constructs the first clusters tier by assigning each non-CH node to a CH according to the RSSI value for the link between them. However, during the steady phase, each member node uses its Time Division Multiple Access (TDMA) schedule like on LEACH to transmit its data to the next hop. When a simple node finishes its data transmission slot, it enters the sleep state to save its energy. PSO-HC is compared to clustering algorithms LEACH, LEACH-C, and PSO-C. The obtained results show that the average energy consumption per node of PSO-HC is less than the other algorithms. The results also reveal that the link quality when using PSO-HC is better than the other because it has the highest throughput among all.

##### MPSICA

MPSICA is an intelligent and fast routing recovery scheme based on PSO [58]. It was performed for heterogeneous WSN and should maintain K disjoint paths from each source node to the nearest super-node (CH) and the available path from super-nodes to the sink. MPSICA is highly fault tolerant to the failure of ordinary node paths or CHs paths, since traditional retransmissions can be decreased and the reliability can be improved with lower energy consumption and longer lifetime. In order to efficiently use the energy consumption, MPSICA is centralized and is performed by each of the super-nodes which have higher resources than simple nodes. Each node on MPSICA represents a particle where some nodes can form a particle sequence corresponding to a path until the sink. The proposed algorithm then optimizes for each node’s particle sequence the optimal path by using an optimal fitness function on the super-node. MPSICA is compared to the clustering algorithm EEHC and the fault tolerant protocol for Inter-Cluster Communication (ICE). The results obtained have shown that MPSICA has the minimum ratio of energy depletion (the higher ratio of alive nodes) compared to the other two protocols. The results also showed that the proposed protocol presents a smaller delay of packet delivery.

##### TPSO-CR

Elhabyan & Yagoub [59], proposed a two-tier PSO protocol for the Two-tier Particle Swarm Optimization for Clustering and Routing protocol (TPSO-CR) in WSNs. The proposed algorithm used a clustering algorithm which finds the optimal set of CHs in order to efficiently use the energy, and maximize the transmission reliability and the network coverage. Meanwhile, its routing algorithm is developed with a novel particle encoding scheme and fitness function which finds the optimal routing tree that connects these CHs to the BS. TPSO-CR is similar to PSO-HC in the sense that it is centralized and based on the information received by the BS. After the computation of the average energy level of nodes by the BS, only nodes with an energy level above the average are eligible to be CCHs for each round to ensure that only nodes with sufficient energy are selected as CHs. Next, the BS runs the first-tier algorithm (clustering algorithm) which find the best K-CHs. The second-tier algorithm (routing algorithm) is performed after the first-tier and is responsible for constructing the optimal routing tree. In order to evaluate TPSO-CR, the authors have used two different scenarios WSN#1 and WSN#2 which use, respectively, homogeneous and heterogeneous nodes. The PSO-based approach is compared with the well-known protocols LEACH, EHE-LEACH, EEHC, LEACH-C, PSO-C, and GA-C. The results have shown that TPSO-CR has a smaller number of non-clustered nodes than other protocols on both scenarios WSN#1 and WSN#2. The results have also revealed that TPSO-CR has the highest throughput, its energy consumption is less than energy consumption of LEACH, EEHC, EHE-LEACH but is slightly similar to the consumed energy of protocols LEACH-C, PSO-C, and GA-C.

##### PSO-ECHS

Srinivasa Rao et al. [60] proposed an energy efficient CH selection algorithm based on PSO called PSO-ECHS. It consists in two phases: the CH election and the cluster formation. The CH election phase is based on PSO and considers the residual energy of nodes and their distance with the BS. Initially, nodes send their location and residual energy to the BS which executes the PSO algorithm. For the CH selection in PSO-ECHS, a particle represents optimal positions of the CHs by representing the coordinates of the sensor nodes to be selected as CH. Contrary to algorithms where non-CH sensor nodes simply join a cluster by considering only the distance with the CH, in PSO-ECHS, the non-CH sensor nodes join a CH depending on the weight function called CH_Weight which consider the distance, the remaining energy, and the node degree of CHs. The evaluation of PSO-ECHS consists of three different scenarios on the position of the BS: at the first scenario, the BS is located in the center of the study field; the second scenario considers the BS located at the top right corner of the field; and the last scenario assumes the BS is located outside the field. Experimentations are conducted on these scenarios by varying the number of nodes on four networks (300, 400, 500, and 700). The proposed algorithm is compared to LEACH, E-LEACH, LEACH-C, PSO-C, and the Least Distance Clustering algorithm (LDC) [61]. The results show that PSO-ECHS outperforms the others algorithms in terms of the energy consumption and thereafter has the best network lifetime. The results have also shown that PSO-ECHS had the highest packets received by the BS in the different scenarios with network which consist of 300, 400, 500, or 700 nodes.

#### 3.5.2. Ant Colony Optimization

The Ant Colony Optimization (ACO) is defined by [62] as a novel nature-inspired metaheuristic for the solution of hard combinatorial optimization problems. The ACO algorithm originates from the behavior of ants which communicate with each other by using chemical deposits called pheromones. When ants move, they lay pheromones on the ground, and they receive the current strength of pheromone [29]. Figure 6 below presents the simple ACO functioning in which ants find the best path between a food source and the nest. The main idea of the ACO metaheuristic is to model the problem as a search for the best path by constructing a path-graph that represents the states of the problem [9]. Many works in the WSN field used clustering algorithms based on the ACO to improve the performance of sensor networks.

##### T-ANT

T-ANT is a clustered data dissemination strategy based on the ant swarm behavior [63]. It promotes a uniform distribution of CHs, which subsequently enables substantial energy savings. The cluster formation is achieved by exploiting two swarm behaviors: foraging and brood sorting. During the cluster setup (CS) phase, a node checks to see whether or not it possesses an ant when the timer expires. If the node has an ant, it becomes CH and advertises its neighbors by broadcasting a message with its node ID. During the set-up phase, the election of CHs uses a swarm of ants in which an ant corresponds to a control message. The ant could travel into the network up to the sink as deep as restricted by its time-to-live (TTL) field. When an ant arrives at a node, the next node is randomly chosen. In order to evaluate the performance of T-ANT, comparison with The Time-Controlled Clustering Algorithm (TCCA) and a multihop version of LEACH called m-LEACH is achieved. The obtained results have shown that the average energy consumed per round in T-ANT is slightly better the other two protocols. However, concerning the number of alive nodes, T-ANT highly achieves the best number of alive nodes.

##### EBAB

The Energy Balanced Ant-Based Routing Protocol (EBAB) [64] is a new adaptive dynamic routing algorithm based on simple biological ants that explore the network and find routes. EBAB produces clusters of unequal sizes to balance energy consumption. Clusters closer to the BS have smaller cluster sizes in order to maximize the lifetime of the network. According to the signal sent by the BS, all the nodes calculate their distance from the BS based on the RSSI, and then nodes decide to compete if necessary to be the unique CH when they are in the same area. In order to form clusters, each CH sends message in a range to tell the other nodes. The nodes send “ACK” message to join the cluster. However, if the node receives more than one message, it will choose the optimal CH according the distance and the energy. The inter-cluster communications are performed by an ameliorated version of ACO in which CH generates initially a Forward Ants (FA). If the FA is not on the BS, it continues to the next step, nevertheless, if it is on the BS, it then activates the backward ants which will increase the pheromone value. The proposed algorithm EBAB is compared to the traditional clustering LEACH in terms of the network lifetime and the amount of data received by the BS. Results have shown that the lifetime of EBAB (number of alive nodes) is longer than the network of LEACH because of the balanced model of the proposed algorithm. Results have also revealed that the amount of data received by EBAB is 1.7 times higher than LEACH.

##### ACO-C

The authors of [65] proposed an energy aware protocol called Ant Colony Optimization for Clustering (ACO-C). ACO algorithms have been used for clustering problems in which N objects are assigned to K clusters based on the distance between each object and the middle of the cluster. ACO-C is centralized, BS selects only nodes with the residual energy above the average energy of the network as possible CHs and uses the ACO algorithm to find the best solution according to a cost function. After finding the best route, the BS informs each node about its CHs and cluster it belongs to. Each CH has to coordinate the transmission of data in their corresponding clusters by using a simple TDMA schedule. At the end of each round, the CH aggregates data it received from its members, and sends the aggregated data to the BS. The algorithm ACO-C is evaluated in two network scenarios; each one contains 100 nodes with unequal initial energy. On the first scenario the BS is located in the middle of the network whereas on the second scenario it is located outside the sensor field. ACO-C is compared with LEACH, LEACH-C, and PSO-C in both scenarios. The obtained results have shown that ACO-C performs the best number of alive nodes through the time, and thus increases the network lifetime better than the three other algorithms. The issues also reveal that ACO-C outperforms LEACH, LEACH-C, and PSO-C in terms of the number of data received by the BS. 

##### ACA-LEACH

The ACA-LEACH algorithm proposed in [66] is an improved version of LEACH which performs the Ant Colony Algorithm (ACA) into inter-cluster routing mechanism to reduce the energy consumption of CHs and finally prolong the lifetime of networks. ACA-LEACH also considers the node residual energy and the distance between the nodes and CHs for the selection of CHs. When the distance between two CHs is less than the threshold distance, ACA-LEACH compares the energy value between the two, and then chooses the CH with more energy as the new CH but with less as the member node. Due to the random choice of CHs, the phenomenon of relative concentration of CHs may exist. *k* ants are placed on each CH and set a matrix which is used to store and record the generated path, ACA selects the optimal path between each CH and the sink helps by the value of pheromones. Afterwards, the *k* ants in each CH are transferred to the next accessible CH with a possibility *p* of each adjacent CHs being selected as the next hop of adjacent clusters. The *k* ants in each CH should select the shortest path with the lowest energy consumption. In order to evaluate ACA-LEACH, the authors have used 200 nodes randomly deployed in a squared sensor field. ACA-LEACH is compared to traditional LEACH and results have shown that the proposed algorithm has a lower average energy consumption than LEACH and its number of alive nodes is better than in the other algorithm.

##### MRP

MRP is a Multipath Routing Protocol for clustering based on Ant Colony Optimization for WSNs [67]. MRP adopts a dynamic clustering algorithm and is divided into three main phases: cluster formation, multipath construction, and data transmission. During the cluster formation, only a node with higher residual energy, more neighbors and stronger RSSI (more than a threshold) has more opportunity of becoming a CH. An improved version of ACO algorithm is used in order to establish multiple paths with minimal energy cost. Three types of ants are used in MRP: *search ants* (*SANTs*) which have to find paths and gather information along these paths until the sink; *backward ants* (*BANTs*) which are used to evaluate the cost of each path discovered by the *SANTs*; and the *abnormal ants* (*AANTs*), which are used to avoid the failure of the protocol. When the residual energy of the current CH is 50% lower than the average energy of all nodes in the cluster, a new CH will be elected. The CH will initiate a new route discovery process when the number of multiple paths is less than two, which means the reliability of path decreased significantly. The evaluation of MRP is conducted by the used of comparisons metrics like average energy, energy consumption, the deviation of energy and the network lifetime on a network with a population varying from 100 to 500 nodes. Based on these parameters, MRP is compared to the clustering algorithm TEEN and the multipath algorithms MP [68] and MACS [69]. The results have shown that MRP has the best average energy and the lowest energy consumption. The standard deviation of energy in MRP is slightly lower than in TEEN and results have also revealed that the proposed ACO-based improves the network lifetime significantly more than the other three protocols.

#### 3.5.3. Bee Colony Optimization

The Bee Colony Optimization (BCO) protocols are inspired from honeybees foraging behaviors. Insects are capable of individual proactive abilities and self-organizing capacity [13]. Honeybees can be grouped into a colony and living within a hive, and show impressive auto-solving problem capabilities. Scout bees explore the surrounding of the hive in order to detect possible sources of food, when a flower (food) is discovered, the scout bee returns back to the hive to recruit the forager bees through a special dance called waggle dance [70]. The BCO are widely used to solve the clustering NP hard problems efficiently.

##### ABC-C

ABC-C is a novel energy efficient clustering mechanism, based on the Artificial Bee Colony (ABC) algorithm [71]. ABC simulates the intelligent foraging behavior of honeybee swarms and has been successfully implemented in clustering techniques. Three bees’ groups in ABC are performed: onlookers to choose a food source, scouts which perform randomly search, and employed bees which go to the food source visited by another bee. The clustering mechanism of the ABC-C is based on the clustering technique of LEACH protocol where CHs perform data aggregation processes of their clusters. For intra-cluster communication, CHs perform TDMA with the BS. The election of CHs is performed by ABC algorithm at the BS which is a node with unlimited energy supply. Sensor nodes are assigned to the clusters by using the shortest distances until the CHs. After receiving and calculating cross-distance values, the nodes send these values to the BS in order to be used during the selection process. To evaluate ABC-C, the authors considered a network which contains 100 nodes randomly placed. The performance of ABC-C is compared to that of LEACH and PSO-C about the network lifetime (number of alive nodes), amount of received signals and the energy consumption. The results have shown that ABC-C improves the network lifetime (highest number of alive nodes) than the other algorithms. The proposed algorithm also outperforms LEACH and PSO-C in terms of energy use and of the number of received signals.

##### Bee-Sensor-C

Bee-Sensor-C is an energy-aware and scalable multipath routing protocol based on dynamic cluster and foraging behavior of bees swarm [70]. It adopts an enhanced multipath construction strategy so that it achieves the balance of energy consumption within the network, reduces overheads of routing, improves the network scalability and performance, then its lifetime. The event-driven of Bee-Sensor-C allow to perform a dynamic clustering scheme near an event. During the first phase of Bee-Sensor-C (clusters formation), the HiveHeader into the bee-hive claims to be a CH node when it detects an event within an event area. Nodes nearby the event will become activated and measure the specific perceived attribute, then the nodes having information about the event will join the cluster. The second phase (multipath construction) occurs when CH needs to send data until the sink node. The CH looks whether there are appropriate foragers of any nodes in the cluster for the existing multipath. The last phase consists in the transmission and occurs when the backward scout with unique path ID arrives at the CH and recruits foragers using the waggle dance. For evaluated Bee-Sensor-C, two different scenarios are considered and the number of nodes varies until 500 (first scenario). In the first scenario, there is only one event that occurred in the network whereas in the second scenario several events (at least 2) can be detected. The proposed algorithm is compared to the bee-inspired routing protocol BeeSensor [72], IEEABR [73], and the flood forward ant routing FF-Ant [74]. The obtained results have revealed that Bee-Sensor-C consumes less energy but has the highest packet delivery rate compared to the three other protocols.

##### BeeSwarm

Mann et al. [75] proposed a CI based metaheuristic for energy efficient hierarchical routing named BeeSwarm. It is made up of three phases: BeeCluster, BeeSearch, and BeeCarrier. During the set-up phase (BeeCluster), a similar approach of the ABC is used for the formation of optimal clusters known as Bee Zones (BZs) and the selection of a corresponding CH known as Zonal Head (ZH). To form a BZ, BeeSwarm applies an agglomerative hierarchal clustering, in which two clusters separated by a short distance are combined to obtain the optimal cluster in order to minimize the energy consumption. After clustering of nodes, ZHs are elected for each BZ based on the residual energy and the distance between the BS. The BeeSearch phase occurs after the BeeCluster and consists in the discovery of routes by scout bees. The BeeSearch performs the forward-search to explore the network and the backward-search which maintains different routes between BS and nodes. Meanwhile, once nodes are registered to specific ZHs, ZH prepares a TDMA schedule and transmits this schedule to its registered nodes in the cluster, BeeSwarm uses a LP in other to optimize the transmission. The proposed BeeSwarm approach is evaluated and is compared to routing protocols MRP and to the evolutionary algorithm ERP [76] on several performance parameters. The results have shown that BeeWarm slightly outperforms the other routing protocols in terms of the total number of packet delivered and the packet delivery ratio. It is also observed that BeeSwarm consumes less energy than the two other protocols, thus increases the network lifetime.

##### ABC-SD [24]

ABC- SD is a power efficient cluster-based routing algorithm for WSNs. The proposed approach is also based on the ABC algorithm for efficient searching features. ABC-SD is a semi-distributed approach in which the clustering process is achieved at the BS using the residual energy and the position of nodes and their neighbors. Nevertheless, the data routing is realized by all nodes in a distributed manner. During the set-up phase, the BS which is responsible for built clusters and selected their CHs starts by the construction of the set of candidate’s CHs based on position between BS and the residual energy. After the construction of sets, ABC algorithm is executed in order to assign nodes to CH and formed clusters. During the steady-phase a multi-hop TDMA based on a B-MAC which implements a sleep/wake-up mechanism is performed to transmit the gathered data to its corresponding CH, either in a single-hop or in a multi-hop manner, depending on the density of the cluster and the location of the CH. When the clustering process is completed, each cluster member starts a route discovery in order to construct a pre-determined path toward its CH. To evaluate the performance of ABC-SD, authors have compared the proposed algorithm with LEACH, LEACH-C, PSO-C, PSO-HC, and ABC-C. The observations of the results have revealed that ABC-SD has the lowest average energy consumed by nodes and a higher energy efficiency than the other algorithms. It is also observed that ABC-SD minimizes the number of non-clustered nodes, improves the throughput, and maximizes the data delivery compared to the five other approaches. 

## 4. Comparison of the Optimized Clustering Approaches

In this section, we present a summary of some optimized clustering algorithms. They are compared and classified according to the ML/CI used, the data delivery rate, the energy consumption which takes in account the network lifetime, the scalability of the algorithms, the type of the approach which can be centralized or distributed, the integration or not of a data aggregation technique, and the homogeneity of nodes.

The data delivery rate concerns the amount of data received by the BS or a receiver node according to the number of sent data by another node. This metric is very important because it helps to have an overview of the data loss during transfers and reliability during communication.The energy consumption metric here is the most important parameter to take into account to evaluate a clustering algorithm. The energy consumed by a sensor node across the time is strongly related to its lifetime that can affect to the lifetime of the whole network. Then a node with a lower energy consumption would have a longer lifetime than another node consuming more energy.Since our study is focused on large scale WSNs, we examine the scalability of the presented algorithm. The scalability here aims to classify the proposed solutions by the number of nodes present in the sensor field. Algorithms with more than 500 nodes are more interesting for our study because they manage more efficiently the amount of communication between nodes than other with less than 100 or 200 nodes within the sensor field.Approaches used to implement clustering algorithm can be either centralized or distributed. On centralized clustering approaches, important decision making and most of the operations are done at the BS, which is not usually limited by its resources. On these approaches, the MI/CI is implemented by the BS, each node has to send their necessary information (its location, remaining energy, etc.) for computations by the BS. In distributed approaches, the decision and the computation are made by the nodes themselves. The BS receive results of computations or decisions, in some case the BS is replaced by the sink node.Another metric considered in this study is the homogeneity of nodes within the field. The network is said to be homogeneous when all the nodes have the same characteristics (performance of the microcontroller, available memory, communication range, energy level, etc.); however, it is heterogeneous when the algorithm considers one or more special nodes with extended performance compared to other normal nodes. In this last case, these special nodes are mostly set as CHs due to their performance and the number of clusters mostly depend of the number of these special nodes.In order to evaluate the amount of energy consume by each node within the field, clustering approach integrate a radio model which has to be as accurate as possible to real behavior. This parameter is important because most of the energy dissipated by a sensor is caused by transmitting and receiving data through the transceiver. The type of radio model also allows you to know whether or not the implementation in real environment can be similar to the simulation context.For our analysis, we evaluate if the presented algorithms are based or not on multihop communication between nodes. The multihop ability assumes that a node can send a data to another far node out of reach but helps by an intermediate node within its communication range.To consider the fault tolerance of the presented algorithms, we set the multipath metric. Algorithms that implement multipath are more fault tolerance than others that do not use it, if a route to a BS is not available anymore, it is possible to access the BS by a secondary path. This parameter allows an algorithm to avoid the entire network failure caused by nodes crash.

Table 1 presents the crosstab between the ML/CI and the energy consumption. We observe that the approaches which consume the most the energy are based on FL or GA. However, the solutions based on the BCO, PSO, and FL have a lower energy consumption than the others paradigms. Thus, we may say that algorithms based on SI and on FL consume less energy that GA, NN, and RL based algorithms. This can be explained knowing that clustering algorithms based on GA and NN are mostly complex to implement and need more node computations. Due to their limited resources, nodes would waste their energy faster. 

Table 2 compare the scalability of the presented algorithms by the CI-ML used. We note that the optimized strategies based on FL, NN, or GA have less scalability than the others do. However, solutions which used SI are the best in terms of the scalability. The higher scalability of the SI based approaches is due to the fact that the SI considers the collaboration between a large amount of agents in a population. However, on FL, NN, and GA when the number of node increases, the complexity of algorithms increases, the computations made by the nodes or the BS require more time and resources (microcontroller, memory, etc.).

The data delivery rate which fixes the amount of data received by the BS is an important parameter. From Table 3 below, we observe optimized clustering approaches based on BCO, ACO, and PSO have the better data delivery rate than the others CI, it proofs then that solutions based on SI are more adapt for applications design on the amount of data to be delivered at the BS. This difference of data delivery rate can show how the approaches based on the GA, NN, or RL lose data than SI based paradigms which can be considered as more reliable.

Since the main advantage of using the clustering approach is its high scalability, however, it varies in function of the CI approaches.

Table 4, Table 5 and Table 6 present respectively the comparison between the approaches which can be either centralized or distributed in terms of the energy consumption, the data delivery rate, and the scalability. From these tables, optimized clustering which has a centralized approach outperform solutions with a distributed skeleton. Most of the centralized optimized clustering algorithms consume less energy than the distributed strategies. However, for an average consumption, both have approximately the same energy consumption. Centralized algorithms outperform distributed algorithms because the major part of the operations are done at the BS in centralized approaches, each node within the sensor field usually sends only the necessary elements (its position, remaining energy, sensed data, etc.) to the BS most of the time without any additional computing.

From our analysis, we observe that must of the presented algorithms uses the first order as radio/energy model. The first order radio model is widely used because it is simple to implement and considers only the energy spent during the transmission and the reception per bit. The energy consumed during the transmission of *k*-bit over a distance *d* is presented in (1), where $e_{t}$ is the energy dissipated by a bit during its transmission and $\left(e_{d}\times d^{2}\right)$ the energy consumed for transmitting a single bit over a distance *d*.
(1)$$E_{tx}\left(k,d\right)=\left(e_{t}+e_{d}.d^{2}\right).k$$

The energy required for successfully receive *k*-bit is given in (2), where $e_{r}$ represents the energy needed to receive a single bit
(2)$$E_{rx}\left(k\right)=e_{r}.k$$

Other algorithms used either the radio model based on the real transceiver CC2420. According to this choice, we can assume that clustering algorithms which implement the CC2420 radio model (data transfer rate, path loss, energy consumption during transmitting and receiving of data) are more accurate for a real environment deployment. However, the used of first order or CC2420 models are only suitable for terrestrial WSN applications, if the sensor field is different from the ground surface, these radio models will not be appropriate.

Table 7 presents the summary of the comparison between the presented optimized clustering approaches.

## 5. Conclusions and Future Work

In this paper, we conducted a wide review of the recent hierarchical approaches based on CI or ML. To achieve this, we classified these algorithms in terms of the CI used which can be FL, GA, NN, RL or SI. In order to evaluate and to compare them, we consider 10 parameters: CI used; the data delivery rate (depend of the amount of data received by the BS); data aggregation; the energy consumption that characterize the network lifetime; the scalability of the algorithm when increasing the number of nodes; the approach of the algorithm which can be centralized or either distributed; homogeneity or the heterogeneity of the network that considers whether sensors have or not the same performance; the radio model used by the optimized algorithms which represent the model of energy; multihop to identify if an optimized solution considers or not multihop communications; multipath for the fault tolerance of algorithms. The tables of comparison show that optimized clustering based on SI are more recent and outperforms the other algorithms in terms of scalability, energy consumption and the amount of data delivered to the BS.

As future works, we plan to classify the optimized clustering in terms of WSN types. We would talk about clustering parameters used to WSNs which locations are non-terrestrial or not above the ground but rather under the water or under the ground. In addition to future work, we would discuss about the spectrum-awareness or spectrum-efficient clustering studies in practical large wireless sensor networks.

## Figures and Tables

**Figure 1 sensors-19-00322-f001:**
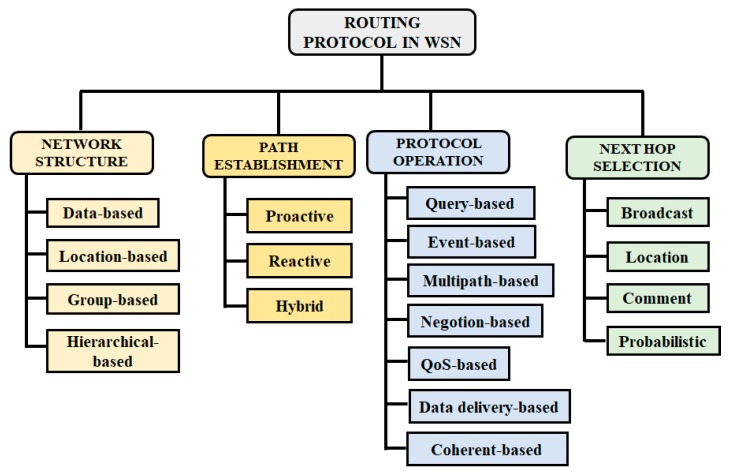
Classification of routing protocol in a Wireless Sensor Network (WSN).

**Figure 2 sensors-19-00322-f002:**
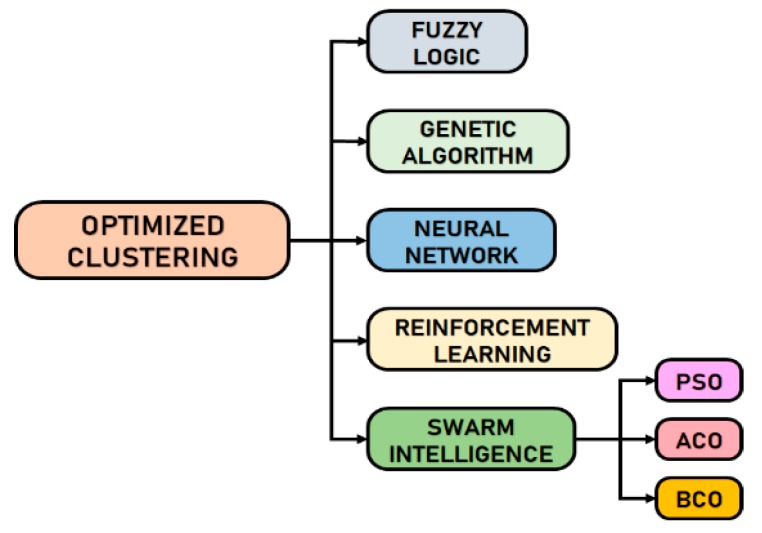
Classification of optimized clustering algorithms according to the Computational Intelligence (CI).

**Figure 3 sensors-19-00322-f003:**
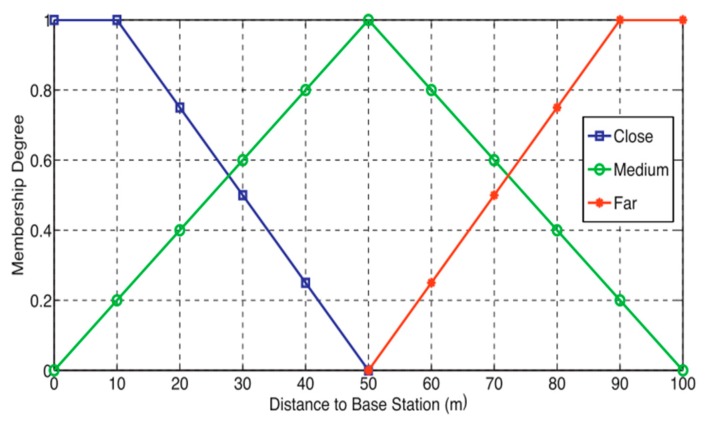
Fuzzy set for input variables of the distance between node and the BS (40).

**Figure 4 sensors-19-00322-f004:**
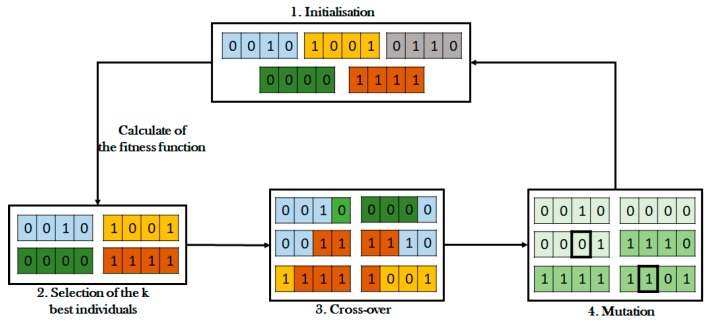
Presentation of the Genetic Algorithm model.

**Figure 5 sensors-19-00322-f005:**
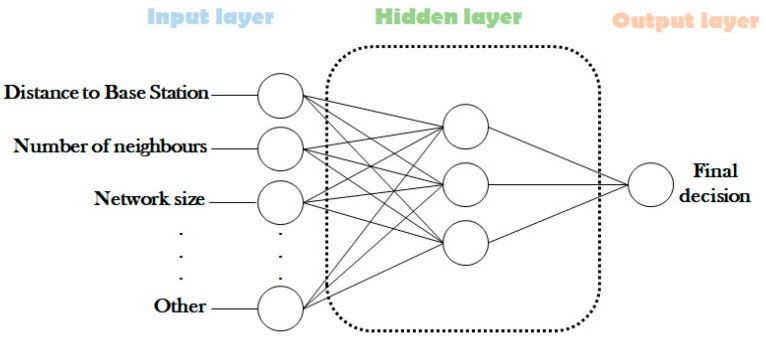
A simple Neural Network model.

**Figure 6 sensors-19-00322-f006:**
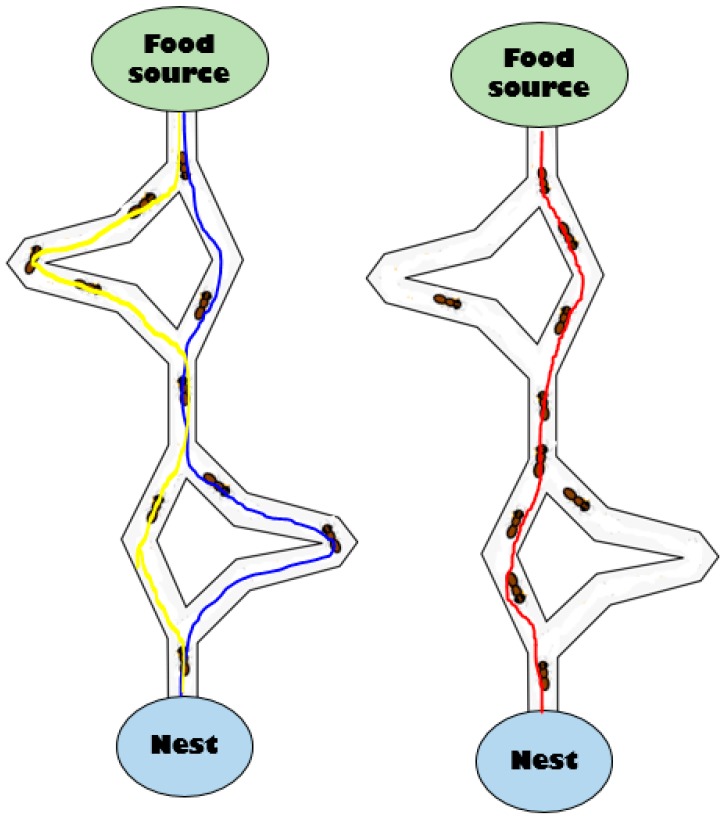
Presentation of the Ant Colony Optimization (ACO) model.

**Table 1 sensors-19-00322-t001:** Comparison of CI’s approaches according to the energy consumption.

ML/CI	Energy Consumption			Total
	Low	Average	High	
FL	3	3	3	9
FL/SI	1	0	0	1
GA	1	2	1	4
NN	0	2	0	2
RL	1	1	0	2
PSO	1	4	1	6
ACO	0	4	1	5
BCO	1	3	0	4
**Total**	**8**	**19**	**6**	**33**

**Table 2 sensors-19-00322-t002:** Comparison of CI’s approaches according to the scalability.

ML/CI	Scalability			Total
	Low	Medium	High	
FL	6	3	0	9
FL/SI	0	1	0	1
GA	2	2	0	4
NN	2	0	0	2
RL	1	1	0	2
PSO	1	0	5	6
ACO	2	1	2	5
BCO	0	2	2	4
**Total**	**14**	**10**	**9**	**33**

**Table 3 sensors-19-00322-t003:** Comparison of CI’s approaches according to the data delivery rate.

ML/CI	Data Delivery Rate			Total
	Low	Average	High	
FL	1	1	0	2
FL/SI	0	1	0	1
NN	0	1	0	1
RL	0	1	0	1
PSO	0	3	2	5
ACO	1	0	1	2
BCO	0	0	4	4
**Total**	**2**	**7**	**7**	**16**

**Table 4 sensors-19-00322-t004:** Comparison of the energy consumption according to the nature of the approach.

ML/CI	Energy Consumption			Total
	Low	Average	High	
Centralized	28.6%	57.1%	14.3%	**14**
Distributed	21.1%	57.9%	21.1%	**19**
**Total**	**8**	**19**	**6**	**33**
	**24.2%**	**57.6%**	**18.2%**	**100%**

**Table 5 sensors-19-00322-t005:** Comparison of the data delivery rate according to the nature of the approach.

ML/CI	Data Delivery Rate			Total
	Low	Average	High	
Centralized	0.0%	37.5%	62.5%	**8**
Distributed	25.0%	50.0%	25.0%	**8**
**Total**	**2**	**7**	**7**	**16**
	**12.5%**	**43.8%**	**43.8%**	**100%**

**Table 6 sensors-19-00322-t006:** Comparison of CI’s approaches according to the nature of the approach.

ML/CI	Scalability			Total
	Low	Medium	High	
Centralized	35.7%	28.6%	35.7%	**14**
Distributed	47.4%	31.6%	21.1%	**19**
**Total**	**14**	**10**	**9**	**33**
	**42.4%**	**30.3%**	**27.3%**	**100%**

**Table 7 sensors-19-00322-t007:** Comparison of the optimized clustering algorithms.

	CI/ML	Data Delivery Rate	Data Aggregation	Energy Consumption	Scalability	Nature	Network	Radio Model	Multihop	Multipath
FCH [31]	FL	-	no	high	low	centralized	homogeneous	first order	no	no
CHEF [32]	FL	-	yes	high	medium	distributed	homogeneous	first order	no	-
LEACH-FL [33]	FL	-	-	high	low	centralized	homogeneous	first order	no	yes
ICT2TSK [34]	FL	-	-	low	medium	centralized	homogeneous	first order	no	yes
SEP-FL [35]	FL	-	-	average	low	centralized	heterogeneous	first order	-	-
EAUCF [36]	FL	-	yes	low	low	distributed	homogeneous	first order	yes	yes
DFLC [37]	FL	low	-	average	medium	distributed	homogeneous	first order	yes	yes
SIF [39]	FL/SI	average	no	low	medium	centralized	homogeneous	first order	yes	yes
FBUC [40]	FL	-	-	low	low	distributed	homogeneous	first order	-	yes
EEDCF [41]	FL	average	-	average	low	distributed	heterogeneous	first order	yes	yes
[43]	GA	-	no	average	medium	centralized	heterogeneous	first order	yes	-
[44]	GA	-	yes	average	medium	distributed	homogeneous	first order	-	-
LEACH-GA [46]	GA	-	yes	high	low	distributed	homogeneous	first-order	no	yes
GABEEC [46]	GA	-	yes	low	low	distributed	homogeneous	first order	-	-
[48]	NN	average	yes	average	low	distributed	homogeneous	-	-	-
[49]	NN	-	-	average	low	distributed	homogeneous	first order	yes	yes
CLIQUE [50]	RL	average	yes	low	medium	distributed	homogeneous	-	yes	yes
[51]	RL	-	-	average	low	distributed	homogeneous	-	-	-
PSO-C [57]	PSO	average	yes	average	low	centralized	homogeneous	first order	no	no
[56]	PSO	average	yes	average	high	centralized	heterogeneous	first order	yes	yes
PSO-HC [57]	PSO	-	-	average	high	centralized	homogeneous	CC2420	yes	-
MPSICA [58]	PSO	average	yes	high	high	distributed	heterogeneous	-	yes	yes
TPSO-CR [59]	PSO	high	yes	average	high	centralized	homogeneous/ heterogeneous	CC2420	yes	yes
PSO-ECHS [60]	PSO	high	no	low	high	centralized	homogeneous	first order	-	-
T-ANT [63]	ACO	-	yes	average	low	distributed	homogeneous	first order	yes	-
EBAB [64]	ACO	low	-	average	high	distributed	homogeneous	first order	yes	yes
ACO-C [65]	ACO	high	yes	average	low	centralized	homogeneous	first order	no	yes
ACA-LEACH [66]	ACO	-	-	high	medium	distributed	homogeneous	first order	yes	yes
MRP [67]	ACO	-	-	average	high	distributed	homogeneous	first order	yes	yes
ABC-C [71]	BCO	high	-	average	medium	centralized	homogeneous	first order	yes (2 hops)	yes
Bee-Sensor-C [70]	BCO	high	yes	average	high	distributed	homogeneous	-	yes	yes
BeeSwarm [75]	BCO	high	yes	average	medium	distributed	homogeneous	-	yes	yes
ABC-SD [24]	BCO	high	yes	low	high	centralized	homogeneous	CC2420	yes	yes

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
