# Peer review of "Optimized Clustering Algorithms for Large Wireless Sensor Networks: A Review"

_sensors, 2019, doi:10.3390/s19020322_

Round 1

Reviewer 1 Report

This survey paper is comprehensive and informative for general readers. The writing and organization are good to follow. Overall, I am positive with this work. Several minor issues are as follows:

Comparison among existing algorithms in term sof the network scale is of great interest and should be discuss in more detail since the titile is "Optimized Clustering Algorithms for Large Wireless 2 Sensor Networks: A Review."

Compared with centralized clustering approaches, more discussions on decentralized clustering are expected by refering exisitng work like: Decentralized sensor selection for cooperative spectrum sensing based on unsupervised learning

Spectrum-awareness or spectrum-efficient clustering studies should be discussed since in practical large wireless sensor networks, the use of spectrum is a vital issue: 

Spectrum Sensing in Opportunity-Heterogeneous Cognitive Sensor Networks: How to Cooperate?

Author Response

Response to Reviewer 1 Comments

Dear reviewer,

thank you very much for your time and efforts reviewing our manuscript. We believe your comment have indeed improved our work and we hope to have answered all open questions and corrected all errors. In the next paragraphs, we answer all comments individually, where the original comment is in black, while our answers are in red for better readability.

Regards,

Damien WOHWE SAMBO,

Blaise Omer YENKE,

Anna FÖRSTER,

Paul DAYANG.

Point 1: Comparison among existing algorithms in terms of the network scale is of great interest and should be discuss in more detail since the title is "Optimized Clustering Algorithms for Large Wireless Sensor Networks: A Review."

Response 1: Thank you for your relevant remark, more details have been added. See section 4.

Point 2: Compared with centralized clustering approaches, more discussions on decentralized clustering are expected by referring existing work like: Decentralized sensor selection for cooperative spectrum sensing based on unsupervised learning.

Response 2:Thank you very much for pointing out this issue. Indeed, there is a big difference between centralized and decentralized clustering approaches, resulting in significant differences in the performance of the network. Generally speaking, it depends on the chosen technique, e.g. Genetic Algorithms are very processing intensive and thus are usually implemented as centralized solutions. Table 7 includes this information in column “Nature”. The mentioned paper has been added also to our survey in Introduction (highlight reference 6).

Point 3: Spectrum-awareness or spectrum-efficient clustering studies should be discussed since in practical large wireless sensor networks, the use of spectrum is a vital issue: Spectrum Sensing in Opportunity-Heterogeneous Cognitive Sensor Networks: How to Cooperate?

Response 3:This is of course an interesting topic, but we consider it out of scope for our paper, which is already quite long and broad. We suggest this as a possible extension of our work.

p { margin-bottom: 0.25cm; direction: ltr; line-height: 115%; text-align: left; }p.western { font-size: 12pt; }p.cjk { font-size: 12pt; }

Reviewer 2 Report

This paper is a reviewing work for the clustering on Wireless Sensor Networks, mainly focus on machine learning and Computational Intelligence. Even it looks like there are lots of related papers reading and comparison work, I cannot follow the meaning of this research very well. The clustering method for WSN should be varied and closely depend on the different applications, after these comparing in this paper, I still cannot find sufficient information and evidential suggestion for algorithms choosing. So, what is the significance of this reviewing study?

Author Response

Response to Reviewer 2 Comments

Dear reviewer,

thank you very much for your time and efforts reviewing our manuscript. We believe your comment have indeed improved our work and we hope to have answered all open questions and corrected all errors. In the next paragraph, we answer all comments individually, where the original comment is in black, while our answers are in red for better readability.

Regards,

Damien WOHWE SAMBO,

Blaise Omer YENKE,

Anna FÖRSTER,

Paul DAYANG.

Point 1: This paper is a reviewing work for the clustering on Wireless Sensor Networks, mainly focus on machine learning and Computational Intelligence. Even it looks like there are lots of related papers reading and comparison work, I cannot follow the meaning of this research very well. The clustering method for WSN should be varied and closely depend on the different applications, after these comparing in this paper, I still cannot find sufficient information and evidential suggestion for algorithms choosing. So, what is the significance of this reviewing study?

Response 1: Thank you for relevant remark, this study aims at giving a quick overview to beginner or senior in the research’s field about clustering approaches based on ML and CI. Readers can find elements for choosing the suitable ML – CI paradigm for its clustering needs according to the specifications of its application. More details have been added in Abstract, Introduction and in the Section 4.

p { margin-bottom: 0.25cm; direction: ltr; color: rgb(0, 0, 0); line-height: 115%; text-align: left; }p.western { font-family: "Calibri", serif; font-size: 12pt; }p.cjk { font-family: "SimSun"; font-size: 12pt; }p.ctl { font-size: 12pt; }

Reviewer 3 Report

- In the introduction, this paper describes well the previous works. However, the object and subject of this paper are a little bit ambiguous. I hope the authors should add a more detailed and explicit purpose of this paper.

- If possible, the authors should mention about definite advantage and disadvantage for cited papers in previous works.

- A revision of English can improve the quality of the article.

Author Response

Response to Reviewer 3 Comments

Dear reviewer,

thank you very much for your time and efforts reviewing our manuscript. We believe your comment have indeed improved our work and we hope to have answered all open questions and corrected all errors. In the next paragraphs, we answer all comments individually, where the original comment is in black, while our answers are in red for better readability.

Regards,

Damien WOHWE SAMBO,

Blaise Omer YENKE,

Anna FÖRSTER,

Paul DAYANG.

Point 1: In the introduction, this paper describes well the previous works. However, the object and subject of this paper are a little bit ambiguous. I hope the authors should add a more detailed and explicit purpose of this paper.

Response 1: Thank you very much for pointing out this issue, we have adapted the Abstract and Introduction to better explain our scope and goals.

Point 2: If possible, the authors should mention about definite advantage and disadvantage for cited papers in previous works.

Response 2:Thank you for the proposition. It can be noticed that the choice of the ML-CI approaches is related to the application’s requirements. We therefore added comments and details for more explanation. See Section 4.

Point 3: - A revision of English can improve the quality of the article.

Response 3:Done.

p { margin-bottom: 0.25cm; direction: ltr; color: rgb(0, 0, 0); line-height: 115%; text-align: left; }p.western { font-family: "Calibri", serif; font-size: 12pt; }p.cjk { font-family: "SimSun"; font-size: 12pt; }p.ctl { font-size: 12pt; }

Reviewer 4 Report

Wireless Sensor Networks are exhaustively surveyed and an interesting and useful comparison is presented. The authors conclude that the best optimizing techniques for clustering optimization are of heuristic type.

I would recommend to change the headings at the lines numbered 299 311 349 363 431. It is rather strange to use biblio cites as headings.

In lines 730 733 Eqs. (1) and (2) should be formatted in mathematical mode.

Author Response

Response to Reviewer 4 Comments

Dear reviewer,

thank you very much for your time and efforts reviewing our manuscript. We believe your comment have indeed improved our work and we hope to have answered all open questions and corrected all errors. In the next paragraphs, we answer all comments individually, where the original comment is in black, while our answers are in red for better readability.

Regards,

Damien WOHWE SAMBO,

Blaise Omer YENKE,

Anna FÖRSTER,

Paul DAYANG.

Point 1: I would recommend to change the headings at the lines numbered 299 311 349 363 431. It is rather strange to use biblio cites as headings.

Response 1: Thank you for this suggestion. We would like to do as you mentioned but the authors themselves do not particularly named algorithms in their paper, it is not therefore appropriate for us the allocate a random name without author’s permissions.

Point 2: In lines 730 733 Eqs. (1) and (2) should be formatted in mathematical mode.

Response 2:Done, see lines 788 and 791.

p { margin-bottom: 0.25cm; direction: ltr; color: rgb(0, 0, 0); line-height: 115%; text-align: left; }p.western { font-family: "Calibri", serif; font-size: 12pt; }p.cjk { font-family: "SimSun"; font-size: 12pt; }p.ctl { font-size: 12pt; }

Reviewer 5 Report

This paper presents a survey of clustering algorithms for wireless sensor networks (WSNs). Since WSN nodes are clustered for energy saving and efficient routing, the authors also introduce some routing schemes for WSNs. Two types of clustering algorithms, CI and ML, are introduced. The authors also provide qualitative and quantitative comparisons for different algorithms. 

The writing of this paper is acceptable. However, the organization can be improved in several aspects. First, the authors do not discuss the overhead of information acquirement for clustering algorithms. For example, the authors show that some algorithms are centralized in Table 7, but there is no discussion about the prerequisite. It would be helpful to consider the messaging overhead. Each metric in Table 7 should be defined. Second, it is not clear how the authors get the numerical results in most tables. If the authors implement different algorithms, please clearly state it. I also feel that the description to the tables in Section 4 is over-simplified. Third, the authors classify clustering algorithms into five types of fuzzy logic, genetic algorithm, neural network reinforcement learning, and swarm intelligence. There are still some algorithms which are based on some deterministic properties in a distributed manner. The authors should consider the categorization of these algorithms. 

Author Response

Response to Reviewer 5 Comments

Dear reviewer,

thank you very much for your time and efforts reviewing our manuscript. We believe your comment have indeed improved our work and we hope to have answered all open questions and corrected all errors. In the next paragraphs, we answer all comments individually, where the original comment is in black, while our answers are in red for better readability.

Regards,

Damien WOHWE SAMBO,

Blaise Omer YENKE,

Anna FÖRSTER,

Paul DAYANG.

Point 1: The writing of this paper is acceptable. However, the organization can be improved in several aspects. First, the authors do not discuss the overhead of information acquirement for clustering algorithms. For example, the authors show that some algorithms are centralized in Table 7, but there is no discussion about the prerequisite. It would be helpful to consider the messaging overhead. Each metric in Table 7 should be defined. 

Response 1: Thank you very much for pointing out this issue, the metric used in Table 7 have been defined and some details have been added. See Section 4.

As for the overhead evaluation, we fully agree with the reviewer that that would be a very valuable addition. In general, and as we explain in Section 4, overhead of centralized approaches is higher. However, exact numbers are almost impossible to acquire. They depend on used hardware, protocols, operating systems, lifetime of the network, etc. Any attempt to quantify the overhead of individual protocols would lead us to guessing them and thus to a unfair comparison.

Point 2: Second, it is not clear how the authors get the numerical results in most tables. If the authors implement different algorithms, please clearly state it. I also feel that the description to the tables in Section 4 is over-simplified. 

Response 2:Thank you, the numerical results in the tables are obtained after a descriptive statistical analysis with IBM SPSS software. Due to the qualitative nature of our parameters, we have opted for cross-tabs which are suitable for this case. Additional comments have been provided in Section 4 in order to explain the information presented in each table.

Point 3: Third, the authors classify clustering algorithms into five types of fuzzy logic, genetic algorithm, neural network reinforcement learning, and swarm intelligence. There are still some algorithms, which are based on some deterministic properties in a distributed manner. The authors should consider the categorization of these algorithms.

Response 3:Thank you for your remarks, however, it is important to note that the list of algorithms we proposed here is not exhaustive. The main objective of our study was to compare and evaluate clustering algorithms based on the five types mentioned in the document. It is also important to note that there are other paradigms used in clustering solutions (for example Firefly algorithm based clustering technique for Wireless Sensor Networks), we focused on the most ML-CI used for clustering solution whether they are centralized or distributed.

p { margin-bottom: 0.25cm; direction: ltr; color: rgb(0, 0, 0); line-height: 115%; text-align: left; }p.western { font-family: "Calibri", serif; font-size: 12pt; }p.cjk { font-family: "SimSun"; font-size: 12pt; }p.ctl { font-size: 12pt; }

Round 2

Reviewer 2 Report

After carefully this revised paper reading, I cannot be persuaded that it has been significantly improved. As a reviewing paper, the most important and difficult work is that the depth of surveying should be helpful for other scholars’ research. However, this paper is lack of valuable comments because of the missing of applications background supporting. In my opinion, reviewing paper is better for those who had finished at least some related researches.

Author Response

Dear reviewer,

thank you very much for your time and your insightful comments and remarks that lead to improvement of the content and presentation of the paper. Please find below your comment (in black) and the corresponding answer (in red).

Point: After carefully this revised paper reading, I cannot be persuaded that it has been significantly improved. As a reviewing paper, the most important and difficult work is that the depth of surveying should be helpful for other scholars’ research. However, this paper is lack of valuable comments because of the missing of applications background supporting. In my opinion, reviewing paper is better for those who had finished at least some related researches.

Response : We would like to point out that we have conducted a wide review of existing solutions in the studied field. We are not yet applying clustering algorithms to some specified fields which is indeed considered in future work, once the classification has been done according to our 10 parameters.

Regards,

Damien WOHWE SAMBO,

Blaise Omer YENKE,

Anna FÖRSTER,

Paul DAYANG.

Reviewer 3 Report

The authors have addressed my comments. In my opinion, the paper can be accepted.

Author Response

Dear reviewer,

thank you very much for your time and your insightful comments and remarks that lead to improvement of the content and presentation of the paper.

Regards,

Damien WOHWE SAMBO,

Blaise Omer YENKE,

Anna FÖRSTER,

Paul DAYANG.